# Excited-State Polarizabilities: A Combined Density Functional Theory and Information-Theoretic Approach Study

**DOI:** 10.3390/molecules28062576

**Published:** 2023-03-12

**Authors:** Dongbo Zhao, Xin He, Paul W. Ayers, Shubin Liu

**Affiliations:** 1Institute of Biomedical Research, Yunnan University, Kunming 650500, China; 2Qingdao Institute for Theoretical and Computational Sciences, Shandong University, Qingdao 266237, China; 3Department of Chemistry and Chemical Biology, McMaster University, Hamilton, ON L8S 4M1, Canada; 4Research Computing Center, University of North Carolina, Chapel Hill, NC 27599-3420, USA; 5Department of Chemistry, University of North Carolina, Chapel Hill, NC 27599-3290, USA

**Keywords:** density functional theory, information theory, excited-state polarizability, ESIPT (excited-state intramolecular proton transfer)

## Abstract

Accurate and efficient determination of excited-state polarizabilities (*α*) is an open problem both experimentally and computationally. Following our previous work, (Phys. Chem. Chem. Phys. 2023, 25, 2131−2141), in which we employed simple ground-state (S_0_) density-related functions from the information-theoretic approach (ITA) to accurately and efficiently evaluate the macromolecular polarizabilities, in this work we aimed to predict the lowest excited-state (S_1_) polarizabilities. The philosophy is to use density-based functions to depict excited-state polarizabilities. As a proof-of-principle application, employing 2-(2′-hydroxyphenyl)benzimidazole (HBI), its substituents, and some other commonly used ESIPT (excited-state intramolecular proton transfer) fluorophores as model systems, we verified that either with S_0_ or S_1_ densities as an input, ITA quantities can be strongly correlated with the excited-state polarizabilities. When transition densities are considered, both S_0_ and S_1_ polarizabilities are in good relationships with some ITA quantities. The transferability of the linear regression model is further verified for a series of molecules with little or no similarity to those molecules in the training set. Furthermore, the excitation energies can be predicted based on multivariant linear regression equations of ITA quantities. This study also found that the nature of both the ground-state and excited-state polarizabilities of these species are due to the spatial delocalization of the electron density.

## 1. Introduction

Molecular polarizability, especially static electric dipole polarizability (α), is a fundamental physicochemical property. It reflects the change of a molecule’s dipole moment in a linear-response manner resulting from an external electric field perturbation [1]. It is a fundamental dielectric property important in many disciplines, especially in materials science [2,3]. Experimental determination of excited-state electrostatic properties is mainly determined using Stark spectroscopy, the electronic absorption/emission [4,5] method, and the flash photolysis time-resolved microwave-conductivity (FP-TMRC) [6,7] technique.

In classical physics, polarizability can be approximately obtained in terms of the volume of a system [8,9]. For example, many strong correlations have been observed for both atoms and molecules [10,11,12,13,14,15,16,17,18]. Tkatchenko and Scheffler (TS) [19] proposed use of atomic volumes and atomic polarizabilities to predict the ground-state polarizabilities for small molecules. Recent progress can be found in [20]. However, performance for excited-state systems has not been reported. 

In quantum mechanics, polarizability can be obtained by iteratively solving the coupled-perturbed Hartree–Fock (CPHF) equation [21,22] or its Kohn–Sham DFT (density functional theory) counterpart [23]. This protocol requires a sufficiently large basis set with polarization and diffuse functions and huge computational costs, but the computational barriers can be partly overcome by using some linear-scaling methods [24,25,26]. In addition, machine learning (ML)-based [27,28,29] methods and a regression-based [30] model have been applied to predict the S_0_ polarizabilities. Polarizability can be related to the band gap of HOMO (highest occupied molecular orbital) and LUMO (lowest unoccupied molecular orbital) in an inverse manner [31,32,33]. 

In the literature, only a few studies [34,35,36,37,38,39,40] have reported excited-state polarizabilities. This is likely because accurate predictions of excited-state geometries and molecular properties of large molecules are difficult, especially when there are perturbations such as external fields. 

Following our previous work, used information-theoretic approach (ITA) quantities to predict the S_0_ polarizabilities of both small and large molecules [41,42]. Here, we aim to predict the S_1_ polarizabilities of 2-(2′-hydroxyphenyl)benzimidazole (HBI, **1**) and its derivatives as shown in Figure 1. For **1**, it is well-documented [43] that the S_0_ (Figure 1a) intramolecular proton transfer (IPT) reaction takes place with difficulty, whereas the S_1_ (Figure 1b) or T_1_ (triplet, not shown) intramolecular proton transfer (ESIPT) process can easily happen. Thus, in this work, only S_0_ and S_1_ were considered to reduce computational cost without compromising the results. We have found that with S_0_ or S_1_ electron densities as inputs, ITA quantities had good correlations with the excited-state polarizabilities. When transition densities were considered, both S_0_ and S_1_ polarizabilities had good relationships with ITA quantities. Furthermore, excitation energies could be predicted based on multiple linear regression equations of ITA quantities. For the first time, we applied the ITA quantities to predict excited-state polarizabilities. It is anticipated that this protocol can be readily applied to condensed-phase systems. 

## 2. Results

Shown in Table 1 are the correlation coefficients (R^2^) between the S_0_ polarizabilities (α_iso_) and ITA quantities, molecular volumes, and quadrupole moments, which were obtained at the CAM-B3LYP/6-311+G(d) level for all the molecular systems shown in Figure 1 and Figure 2. The linear regression equations can be found in Appendix A. It is clear from Table 1 that the Ghosh−Berkowitz−Parr (GBP) entropy (S_GBP_), 2nd and 3rd relative Rényi entropy (^r^R_2_ and ^r^R_3_), information gain (I_G_), G_1_, G_2_, and G_3_, and quadrupole moments (Θ_iso_) have strong linear relationships with α_iso_, with R^2^ > 0.8. Of note, the G_3_ data were shown to be strongly correlated with α_iso_ for various systems, among which were 30 planar or quasi-planar bases [41], and 20/40/8000 amino acids, dipeptides and tripeptides [41,42]. The Θ_iso_ values had a good relationship with α_iso_, the theoretical rationale for which can be found in [42]. However, a solid and sound theoretical verification between G_3_ and α_iso_ was lacking. Additionally, molecular volumes had only reasonably good correlation with α_iso_, with R^2^ = 0.705, indicating that it is not a good descriptor of α_iso_. For Shannon entropy and Fisher information, there exist three exceptions for **2**, **8**, and **13**. It was found that these three molecules are Br-containing, indicating that there may be some regions for heavy atoms where the electron density and/or its gradients are numerically inconsistent. Taking the three outliers away, much stronger correlations were found for Shannon entropy (better than Fisher information). Here, the nature of ground-state polarizabilities was due to the spatial delocalization of the electron density. Overall, the strong correlations indicate that our computational results are sound.

Table 2 shows S_1_ polarizabilities, S_1_ ITA quantities, including Shannon entropy (S_S_), Fisher information (I_F_), 2nd and 3rd relative Rényi entropy (^r^R_2_ and ^r^R_3_), G_2_ and G_3_, molecular volumes (Vol), and quadrupole moments (Θ_iso_), obtained at the TD-CAM-B3LYP/6-311+G(d) level. Table 2 also shows the correlation coefficients (R^2^) between the S_1_ polarizabilities and other quantities at S_1_. The linear regression equations can be found in Appendix A. Note that some ITA quantities, well-defined at S_0_, are numerically inconsistent (for example, negative occupancy) at S_1,_ and are thus missing. One can see that G_2_, G_3_, and Vol have good correlations with α_iso_ at S_1_, with R^2^ > 0.8. It is intriguing that at S_1_ and G_3_ still have good correlations with α_iso_. This is the first time such a phenomenon has been observed. However, admittedly, the theoretical foundation lags behind the numerical evidence found in this work. Moreover, we have found that for Vol, the R^2^ value was much larger at S_1_ (0.818) than that at S_0_ (0.705). One possible reason is that the excited-state relaxation expands the volume and polarizability space. Finally, in Columns 3 and 4 of S_S_ and I_F_, **2**, **8**, and **13** seem to have abnormal values compared with the others. They are all Br-containing, indicating that there may be some regions for heavy atoms where density gradients are numerically inconsistent at S_1_. Overall, we showed that excited-state densities and molecular polarizabilities α*_iso_* are mutually entangled. 

After showing that ITA quantities can be correlated with α_iso_ either at S_0_ or S_1_, it is important to ask if one can use ITA quantities at S_0_ to predict α_iso_ at S_1_. The answer is definitely yes. Table 3 shows the correlation coefficients (R^2^) between the α_iso_ values at S_1_ and ITA quantities, molecular volumes, and quadrupole moments at S_0_ introduced in Table 1. More details can be found in Appendix A. The linear regression equations can be found in Appendix A. One can see that the α_iso_@S_0_, I_G_, G_1_, G_2_, and G_3_ have good relationships with α_iso_@S_1_, with R^2^ > 0.8. Moving forward, we ask if one can use the transition density (matrix) as an input for ITA quantities to correlate with α_iso_ either at S_0_ or S_1_. The answer again is yes. Table 4 shows strong correlations between α_iso_ either at S_0_ or S_1_ and ITA quantities with the transition density matrix as an input. The linear regression equations can be found in Appendix A. Except for Shannon entropy (S_S_) and Fisher information (I_F_), the other ITA quantities have reasonably good or strong correlations with α_iso_, either at S_0_ or S_1_, with R^2^ ranging from 0.72 to 0.94. The implication of this is that electron-density-based quantities can be used to predict excited-state properties, such as molecular polarizabilities. 

To further showcase the transferability of the linear regression models, we compared both the S_0_ and S_1_ polarizabilities for all systems in Figure 3, as shown in Table 5 and Table 6, respectively. Since Shannon entropy and Fisher information have exceptions for the linear regressions, they were not used for predicting the polarizabilities of all systems in Figure 3. From Table 5, one can see that, S_GBP_, ^r^R_2_, ^r^R_3_, and G_3_ perform well in reproducing the conventional S_0_ polarizabilities, as evidenced by the mean unsigned errors (MUEs) and mean signed errors (MSEs). The other five quantities show reasonably good predictions. From Table 6, one can observe that the predicted S_1_ polarizabilities are in reasonably good agreement with the conventional data. All the quantities show comparable performance. Taken together, we have shown that the transferability of the linear regression models is convincing, as the systems in Figure 3 show little or low similarity to those in Figure 1 and Figure 2. 

Next, we compared the α_iso_ data (either at S_0_ or S_1_) predicted by the TS formulas with conventional results as a reference, as shown in Table 7. The predicted α_iso_ results are shown in Appendix A. Employing the original Tkatchenko–Scheffler (TS) formula [19] with Becke [44] or Hirshfeld [45] partitions, the α_iso_ data (either at S_0_/S_1_) were either strongly underestimated or overestimated, with MUE(%) up to −26.44/−24.94 and 7.92/8.93, respectively. It was found that a mean value could reduce the MSE(%) to 14.40/15.53. Moreover, with the new TS formula [20], the results were not improved but worsened, as shown in Table 7. Taken together, we found that the TS formulas have room to improve in predicting S_1_ polarizabilities. 

Finally, we found that excitation energies can be predicted in addition to multivariant linear regression equations of ITA quantities. For example, one can use the S_0_ density matrix as an input for ITA quantities to correlate with excitation energies. Similarly, if the transition density matrix is used for ITA quantities, the excitation energies can also be predicted. Based on the two regression equations constructed from all the systems in Figure 1 and Figure 2, we collected predicted excitation energies, and those from conventional calculations, for all the systems in Figure 3, as shown in Table 8. The MUEs and MSEs were −0.84/1.08 eV and −0.76/1.21 eV, respectively, for the excitation energies. This indicates that the inaccuracy of this protocol is comparable to that of underlying approximations of DFT [46,47]. 

## 3. Discussion

To accurately and efficiently predict excited-state polarizabilities is an ongoing issue. Solving standard CPHF/CPKS equations is computationally intensive, and the computational costs can be excessive for macromolecular systems. Other algorithms and models available in the literature are normally concerned with ground-state polarizabilities. Within this context, we applied density-based ITA quantities to correlate with *α*_iso_ at S_0_/S_1_. This was inspired by our previous work on predicting the ground-state polarizabilities for small and macromolecular systems. Our tentative results show that the protocol is a promising theoretical tool. More systems along this line need to be considered to make this protocol more robust and applicable. We have to point out that when the system under study increases in size, the molecular wavefunctions (thus electron density) become difficult to assess and sometimes computationally unfeasible. Under these circumstances, we have to resort to linear-scaling electronic structure methods, such as GEBF (generalized energy-based fragmentation method) [48,49,50,51], where only small subsystems of a few atoms or groups are treated. 

Next, we adopted the TS method, as mentioned previously. We had found previously that, based upon the Hirshfeld or Becke partition scheme, the original TS formula had unsatisfactory performance either by overestimating or underestimating the S_1_ polarizabilities. An apparent reduction of the deviations can be obtained by averaging the two sets of results. The reason behind this is unclear at the moment. From the original formula,
$$\alpha_{mol}^{TS-old}=\sum_{A}\alpha_{A}^{eff}=\sum_{A}\alpha_{A}^{free}\left(\frac{V_{A}^{eff}}{V_{A}^{free}}\right)$$
one can argue that the weights $\left(\frac{V_{A}^{eff}}{V_{A}^{free}}\right)$ may be the root cause of its poor performance, mainly because the atomic polarizabilities $\alpha_{A}^{free}$ are experimentally determined and computationally verified, as summarized in ref [52]. In the same way, a revised TS formula,
$$\alpha_{mol}^{TS-new}=\sum_{A}\alpha_{A}^{eff}=\sum_{A}\alpha_{A}^{free}{\left(\frac{V_{A}^{eff}}{V_{A}^{free}}\right)}^{4/3}$$
improved performance of predicting S_0_ polarizabilities. However, its predicting power was shown to be far from satisfactory for macromolecules. In this work, we further found that both the original and new TS formulas fail to give a satisfactory description of the S_1_ polarizability. This indicates that the two formulas are oversimplified and may be system-dependent. Overall, the volume-based TS equations are inferior to the density-based ITA regression equations. 

## 4. Materials and Methods

### 4.1. Information-Theoretic Approach Quantities 

Shannon entropy *S*_S_ [53] and Fisher information *I*_F_ [54] are two cornerstone quantities in information theory. They are defined as Equations (1) and (2), respectively:(1)$$S_{S}=-\int \rho \left(r\right)ln\rho \left(r\right)dr$$
(2)$$I_{F}=\int \frac{{\left|\nabla \rho (r)\right|}^{2}}{\rho (r)}dr$$
where $\rho \left(r\right)$ is the electron density and ∇ρ(r) is the density gradient. The physical picture of $S_{S}$ and $I_{F}$ is clear; the former measures the spatial delocalization of the electron density and the latter gauges the sharpness or localization of the same. 

Apart from total density, components such as kinetic-energy density can be used to define an ITA quantity. With both electron density and the kinetic energy density, Ghosh, Berkowitz, and Parr developed a formula for entropy (S_GBP_) [55]: (3)$$S_{GBP}=-\int \frac{3}{2}k\rho \left(r\right)\left[c+ln\frac{t(r;\rho )}{t_{TF}(r;\rho )}\right]dr$$
where *t*(**r**; *ρ*) and *t*_TF_(**r**; *ρ*) represent the non-interacting and Thomas–Fermi (TF) kinetic energy density, respectively. Here *k*, *c*, and *c*_K_ are three constants (*k*, the Boltzmann constant, *c* = (5/3) + ln(4π*c*_K_/3), and *c*_K_ = (3/10)(3π^2^)^2/3^). Full integration of kinetic energy density *t*(**r**; *ρ*) leads to the total kinetic energy *T*_S_ via
(4)$$\int t\left(r;\rho \right)dr=T_{S}$$
while *t*(**r**; *ρ*) can be obtained from the canonical orbital densities,
(5)$$t\left(r;\rho \right)=\sum_{i}\frac{1}{8}\frac{\nabla \rho_{i}\cdot \nabla \rho_{i}}{\rho_{i}}-\frac{1}{8}\nabla^{2}\rho$$
and *t*_TF_(**r**; *ρ*) is simply cast in terms of $\rho \left(r\right)$,
(6)$$t_{TF}(r;\rho )=c_{K}\rho^{5/3}\left(r\right)$$

Of note, the kinetic-energy density can differ in its form, and can be used in different contexts [56,57,58,59,60,61,62,63]. However, S_GBP_ satisfies the maximum-entropy requirement from a mathematical viewpoint [55].

Moving forward, some ITA quantities were introduced for chemical reactions. As new reactivity descriptors in conceptual density functional theory (CDFT) [64,65,66,67], one example is relative Rényi entropy [68] of order *n*
(7)$$R_{n}^{r}=\frac{1}{n-1}ln\left[\int \frac{\rho^{n}\left(r\right)}{\rho_{0}^{n-1}\left(r\right)}dr\right]$$

Information gain [69] (also called Kullback−Leibler divergence or relative Shannon entropy) *I*_G_ is expressed as follows,
(8)$$I_{G}=\int \rho \left(r\right)ln\frac{\rho \left(r\right)}{\rho_{0}\left(r\right)}dr$$

In Equations (7) and (8), *ρ*_0_(**r**) and *ρ*(**r**) satisfy the same normalization condition, and *ρ*_0_(**r**) denotes the reference-state density. 

Recently, in [70] one of the present authors proposed three functions *G*_1_, *G*_2_, and *G*_3_, at both atomic and molecular levels. These are defined as below:(9)$$G_{3}=\sum_{A}\int \rho_{A}\left(r\right)[\nabla ln\frac{\rho_{A}\left(r\right)}{\rho_{A}^{0}\left(r\right)}{]}^{2}dr$$
(10)$$G_{1}=\sum_{A}\int {\nabla^{2}\rho }_{A}\left(r\right)\frac{\rho_{A}\left(r\right)}{\rho_{A}^{0}\left(r\right)}dr$$
(11)$$G_{2}=\sum_{A}\int \rho_{A}\left(r\right)\left[\frac{{\nabla^{2}\rho }_{A}\left(r\right)}{\rho_{A}\left(r\right)}-\frac{\nabla^{2}\rho_{A}^{0}\left(r\right)}{\rho_{A}^{0}\left(r\right)}\right]dr$$

Equations (9)–(11) have been theoretically derived, numerically verified, and used in many applications, as in refs [41,42,70,71]. As one of our major achievements over the past decade, we aimed to combine density functional theory and information theory in a seamless manner, since the two theories can have electron density as an input. Our recent progress along this line can be found in two reviews [72,73]. As another prominent example, we have applied the ITA to assess homochirality [74,75].

Finally, the Hirshfeld’s stockholder approach [45,76,77,78,79] is often introduced to partition atoms in a molecule in the literature, as defined in Equation (12),
(12)$$\rho_{A}\left(r\right)=\omega_{A}\left(r\right)\rho \left(r\right)=\frac{\rho_{A}^{0}\left(r\right)(r-R_{A})}{\sum_{B}\rho_{B}^{0}\left(r-R_{A}\right)}\rho \left(r\right)$$

Here, $\rho_{A}\left(r\right)$ is the atomic Hirshfeld density, $\omega_{A}\left(r\right)$ is a sharing function, and $\rho_{B}^{0}\left(r-R_{A}\right)$ is the atomic density of B centered at **R**_A_. The sum over all the free atom densities, typically spherically averaged S_0_ atomic densities, is usually termed the promolecular density. The Stockholder approach is used in the context of ITA because it is also based on information-theoretic arguments. Alternative partitioning schemes include Becke’s fuzzy atom approach [44] and Bader’s zero-flux atoms-in-molecules (AIM) method [80].

### 4.2. Computational Details

All density functional theory (DFT) calculations were performed with the Gaussian 16 [81] package. Default options included ultrafine integration grids and tight self-consistent field convergence, which were adopted to eliminate numerical noises. The ground- and excited-state structural relaxation was carried out at the CAM-B3LYP/6-311+G(d) [82,83] and TD-CAM-B3LYP/6-311+G(d) [82,83,84,85,86] level, respectively, for 36 molecular systems, as shown in Figure 1, Figure 2 and Figure 3. Optimized atomic Cartesian coordinates are supplied in the Appendix A. Harmonic vibrational frequency calculations were executed at the same level, and no imaginary frequencies were observed by direct visual inspection. Isotropic polarizabilities (α_iso_ = (α_xx_ + α_yy_ + α_zz_)/3) and isotropic quadrupole moments (Θ_iso_ = (Θ_xx_ + Θ_yy_ + Θ_zz_)/3), molecular volumes (at 0.001 e/Bohr^3^ contour surface of electronic density), and molecular wavefunctions, were obtained at the CAM-B3LYP/6-311+G(d) level. The Multiwfn 3.8 [87] program was utilized to calculate all ITA quantities at S_0_ and S_1_ using the molecular wavefunction file as an input. Of note, except for the S_0_ state, the keyword “density=current” in Gaussian 16 was used to dump the density matrix in a wavefunction file. The stockholder Hirshfeld partition scheme of atoms in molecules was employed when atomic contributions were concerned. The reference-state density was the neutral atom calculated at the same level of theory as molecules.

## 5. Conclusions

In this work, we extended the information-theoretic approach (ITA) to treat excited-state polarizabilities. With the ground-state, the first excited-state, or the transition density matrix as an input for ITA quantities, strong linear correlations were observed between molecular polarizabilities and ITA quantities. Based upon these regression equations for molecules in the training set, one can further predict the molecular polarizabilities for other systems in the test set. We verified that good accuracy and transferability of the linear regression models can be obtained. In addition, we found that the nature of both ground-state and excited-state polarizabilities is due to the electron delocalization of the electron density. Finally, we revealed that excitation energies can be predicted on top of a multivariant linear regression equation of ITA quantities. We mention in passing that the ITA quantities were applied to ground-state macromolecules in our previous work and excited-state systems in this work. More work along this line is underway for condensed-phase systems, and the results will be presented elsewhere. 

## Data Availability

Data is contained within the article.

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
