# Peer review of "Excited-State Polarizabilities: A Combined Density Functional Theory and Information-Theoretic Approach Study"

_molecules, 2023, doi:10.3390/molecules28062576_

Round 1

Reviewer 1 Report

This contribution is a theoretical investigation of the relationship between a number of molecular/density descriptors (mainly based on the information theoretic approach) and excited state polarizabilities. By considering the substitution of a single molecule, a dataset of 27 points is generated and the correlation between the excited state polarizability and the descriptors are reported. Robust prediction of excited state polarizabilities is an interesting goal with strong links to materials science. However, the manuscript in its current form seems incomplete and publication at this point strikes me as too early. I expand upon this below.

1) The authors demonstrate a correlation between a number of descriptors and the polarizabillity, including on page 5 the statement "Implication of this part is straightforward that electron-density-based quantities can be used to predict the excited-state properties, such as molecular polarizabilities". The authors need to follow through on this and demonstrate with some tests that this is true. How do we go from the descriptors to the polarizability? Otherwise all we have are some slightly vague correlations and no causation.

2) The authors conclude (end of section 3) that "the volume-based [quantities] are inferior to the density-based ITA quantities". This is not proven in the manuscript. There are no ITA based predictions of the polarizabilities.

3) On page 7, the authors report regression-based equations for predicting emission energies from ITA descriptors:

(i) A similar fit for polarizabilites would hugely strengthen this manuscript and address points 1 and 2 above.

(ii) It's not clear where these equations come from, more detail should be provided.

(iii) It appears that the errors reported for these relationships are for the same molecules as they were fit to. A separate test set should be used to demonstrate if the model is transferable to other systems.

4) I find Table 4 impenetrable. Are there actually two header rows, with inconsistent formatting? Which properties/descriptors correspond to which electronic state? (Two states are mentioned in the caption, but then only one polarizability is reported)

5) One of the authors is not listed in the author contributions section. What was their role in the investigation?

Author Response

To referee 1:

Thank you very much for your kind comments and valuable suggestions. Our responses and changes are listed as follows:

This contribution is a theoretical investigation of the relationship between a number of molecular/density descriptors (mainly based on the information theoretic approach) and excited state polarizabilities. By considering the substitution of a single molecule, a dataset of 27 points is generated and the correlation between the excited state polarizability and the descriptors are reported. Robust prediction of excited state polarizabilities is an interesting goal with strong links to materials science. However, the manuscript in its current form seems incomplete and publication at this point strikes me as too early. I expand upon this below.

1) The authors demonstrate a correlation between a number of descriptors and the polarizability, including on page 5 the statement "Implication of this part is straightforward that electron-density-based quantities can be used to predict the excited-state properties, such as molecular polarizabilities". The authors need to follow through on this and demonstrate with some tests that this is true. How do we go from the descriptors to the polarizability? Otherwise all we have are some slightly vague correlations and no causation.

Response and changes: In the revised MS, we have added more systems in Schemes 2 and 3 to verify that the based on the linear regression equations of ITA quantities with molecular polarizabilities, one can further predict the molecular polarizabilities.   

2) The authors conclude (end of section 3) that "the volume-based [quantities] are inferior to the density-based ITA quantities". This is not proven in the manuscript. There are no ITA based predictions of the polarizabilities.

Response and changes: In the revised MS, we have changed the previous statement to the new one: “Overall, the volume-based TS equations are inferior to the density-based ITA regression equations.”

3) On page 7, the authors report regression-based equations for predicting emission energies from ITA descriptors: (i) A similar fit for polarizabilities would hugely strengthen this manuscript and address points 1 and 2 above. (ii) It's not clear where these equations come from, more detail should be provided.

Response and changes: For polarizabilities, we have used linear regression with a single variable. For excitation energies, we have used a multivariant linear regression strategy of ITA quantities. In the revised MS, we have added all the linear regression equations in the supplementary Tables.

(iii) It appears that the errors reported for these relationships are for the same molecules as they were fit to. A separate test set should be used to demonstrate if the model is transferable to other systems.

Response and changes: To make this model transferrable, we enlarged the “training set” (a total of 36 molecules in Schemes 1 and 2) with more representative molecules which can undergo ESIPT. Based on the linear regression equations in the supplementary Tables, we have predicted both the S0 and S1 polarizabilities for another seven molecules in Scheme 3 and compared them with conventional data. To avoid bias, the molecules we have selected show no or little similarity to those in Schemes 1 and 2. The results show that even with a small amount of data, the models show reasonably good transferability. 

4) I find Table 4 impenetrable. Are there actually two header rows, with inconsistent formatting? Which properties/descriptors correspond to which electronic state? (Two states are mentioned in the caption, but then only one polarizability is reported)

Response and changes: In the revised MS, we have made it more clearly. Both the ground-state (αiso@S0) and the first excited-state (αiso@S1) polarizabilities can be correlated with the ITA quantities with the transition density matrix as an input.

5) One of the authors is not listed in the author contributions section. What was their role in the investigation?

Response and changes: Thanks for pointing this out! In the revised MS, we have added the corresponding contributions.

Reviewer 2 Report

In this manuscript, the authors use machine learning to suggest a model for predicting excited-state polarizabilities based on their information-theoretic approach. This study prepares the learned data using proper calculations and determine the prediction model as usual in machine learning-based studies. Though this paper could be worth publishing, it requires major revisions for the publication. Followings are the points that have to be revised:

1. The authors chose only the properties related to kinetic energy and entropy as the learned data. However, it is not explained why they determine these properties are sufficient to express the prediction model for the excited-state polarizabilities. It should be detailed the cause for the sufficiency, though I suppose it may be explained in their previous paper (Ref. 40).

2. Though the properties of the learned data are very similar to each other, the learning results show that Shannon entropy and Fisher information have much less correlation with the excited-state polarizabilities. The authors should discuss the reason for this result by considering the nature of the excited-state polarizabilities.

3. The method of the machine learning is not sufficiently explained. I cannot find even the type of the regression model, the number of the training and test data, the scores of the test, and other tools for the learning. These information are needed to evaluate the reliability of the learning. Furthermore, these learned data should be given in the supporting information.

Author Response

To referee 2:

Thank you very much for your kind comments and valuable suggestions. Our responses and changes are listed as follows:

In this manuscript, the authors use machine learning to suggest a model for predicting excited-state polarizabilities based on their information-theoretic approach. This study prepares the learned data using proper calculations and determine the prediction model as usual in machine learning-based studies. Though this paper could be worth publishing, it requires major revisions for the publication. Followings are the points that have to be revised:

  1. The authors chose only the properties related to kinetic energy and entropy as the learned data. However, it is not explained why they determine these properties are sufficient to express the prediction model for the excited-state polarizabilities. It should be detailed the cause for the sufficiency, though I suppose it may be explained in their previous paper (Ref. 40).

Response and changes: The information-theoretic approach (ITA) quantities, such as Shannon entropy and Fisher information, are simple density-based functions and well-established in the literature for a long time and some (G1, G2, and G3) have been introduced by us recently in 2019. Our previous publications have numerically verified the intercorrelations between the molecular polarizability and ITA quantities. Following this line, we extend previous work to the excited state since this is an underexplored territory. We have found that the excited-state polarizabilities can correlate with both the ground-state and excited-state ITA quantities.

  1. Though the properties of the learned data are very similar to each other, the learning results show that Shannon entropy and Fisher information have much less correlation with the excited-state polarizabilities. The authors should discuss the reason for this result by considering the nature of the excited-state polarizabilities.

Response and changes: Thanks for pointing this out! In the revised MS, we have added some discussions on the nature of the excited-state polarizabilities. Taking the three outliers away, much more stronger correlations can be found for Shannon entropy (than Fisher information), in this sense, the nature of both the ground-state and excited-stated polarizabilities of these species is due to the   electron delocalization of the electron density.   

  1. The method of the machine learning is not sufficiently explained. I cannot find even the type of the regression model, the number of the training and test data, the scores of the test, and other tools for the learning. These information are needed to evaluate the reliability of the learning. Furthermore, these learned data should be given in the supporting information.

Response and changes: In the revised MS, we added all the linear regression equations in the supplementary Tables. Based on these equations, we have predicted both the S0 and S1 polarizabilities for another seven molecules in Scheme 3 and compared them with conventional data.

To avoid bias, the molecules we have selected show no or little similarity to those in Schemes 1 and 2. The results show that even with a small amount of data, the models show reasonably good transferability.  

Round 2

Reviewer 1 Report

In this revised version of the manuscript the authors have considered the reviewers points and have made suitable alterations to improve the manuscript.

Reviewer 2 Report

I think this manuscript is well improved.

So, I recommend the publication.